# Model-Based Optimization of a Fed-Batch Bioreactor for mAb Production Using a Hybridoma Cell Culture

**DOI:** 10.3390/molecules25235648

**Published:** 2020-11-30

**Authors:** Gheorghe Maria

**Affiliations:** 1Department of Chemical and Biochemical Engineering, University Politehnica of Bucharest, Polizu Str. 1-7, P.O. 35-107, 011061 Bucharest, Romania; gmaria99m@hotmail.com; Tel.: +40-744-830-308; 2Romanian Academy, Calea Victoriei, 125, 010071 Bucharest, Romania

**Keywords:** monoclonal antibodies (**mAbs**) maximization, hybridoma cell culture, fed-batch bioreactor dynamics optimization, raw material consumption, time stepwise operating policy

## Abstract

Production of monoclonal antibodies (**mAbs**) is a well-known method used to synthesize a large number of identical antibodies, which are molecules of huge importance in medicine. Due to such reasons, intense efforts have been invested to maximize the **mAbs** production in bioreactors with hybridoma cell cultures. However, the optimal control of such sensitive bioreactors is an engineering problem difficult to solve due to the large number of state-variables with highly nonlinear dynamics, which often translates into a non-convex optimization problem that involves a significant number of decision (control) variables. Based on an adequate kinetic model adopted from the literature, this paper focuses on developing an in-silico (model-based, offline) numerical analysis of a fed-batch bioreactor (**FBR**) with an immobilized hybridoma culture to determine its optimal feeding policy by considering a small number of control variables, thus ensuring maximization of **mAbs** production. The obtained time stepwise optimal feeding policies of **FBR** were proven to obtain better performances than those of simple batch operation (**BR**) for all the verified alternatives in terms of raw material consumption and **mAbs** productivity. Several elements of novelty (i–iv) are pointed out in the “conclusions” section (e.g., considering the continuously added biomass as a control variable during **FBR**).

## 1. Introduction

Over the last decades, there has been a continuous trend in developing increasing numbers of effective cell or enzymatic reactors [1] to industrialize various biosynthesis technologies for producing industrial fine chemicals (including the bio-mimetic emulations, eventually by using Systems Biology tools, among others [2,3]) by using free-suspended or immobilized cell cultures/enzymes [4]. Effective and selective biosyntheses replaced energetically intensive chemical processes [5,6].

Bioreactors with microbial/animal cell cultures used for the production of a lot of products have been developed in simple constructive/operating alternatives as reviewed in Table 1 together with some examples. Complex alternatives are extensively discussed by [7,8]. In spite of their larger volumes, continuously mixing aerated tank reactors, operated in **BR** (batch), or **FBR** (semi-batch) modes, are preferred for processes requiring high oxygen transfers, and rigorous temperature/pH control, as the case of the approached **mAbs** production.

In addition to production capacity optimization, a crucial engineering problem to be solved concerns the development of optimal operating policies based on an available process model derived from on-/offline measurements. The model-based optimal operation of a bioreactor can be applied in two ways: (a) offline, in which an optimal operating policy is determined by using an adequate kinetic model of the previously identified process, and (b) online, using a simplified (often empirical) model and a classic state-parameter estimator, based on online recorded data [9,13,26,27].

Even if the bioprocess kinetics and biomass characteristics are known, solving this engineering problem is not an easy task due to the presence of multiple (often competing) objectives, technological constraints, and a significant degree of uncertainty originating from multiple sources [10,14,15]. All these parametric/model/data uncertainties require a frequent update of the bioprocess model, the optimal operating policies being determined by using both heuristic and deterministic (model-based) optimization rules [28]. Multi-objective criteria, including economic benefits, operating and materials costs, product quality, etc., are used to derive feasible optimal operating/control policies for various bioreactor types [15] by using specific numerical algorithms [11,13,14,29,30]

Due to such multiple reasons, for a given bioprocess, it is difficult to a priori indicate the least costly optimal operating policy of the chosen bioreactor. This problem is case-dependent.

To be consistent, the in-silico derived optimal operation policy of a bioreactor must be based on a simulation model that must include most of the key variables (particularly the biomass viability) in an adequate bioprocess kinetic model. The use of a deterministic (process mechanism-based) model, as the case here, is often preferred due to the physical significance of the terms/parameters, which make their verification possible vs. observations and the literature, even if repeated model updating is often necessary due to the high variability of the bioprocess. Typical objective functions were reviewed by [5,16].

On the other hand, this preliminary in-silico (model-based) analysis is necessary because: (a) New complex bioprocesses that are developed every year, requiring an engineering analysis to correctly determine the most profitable bioreactor operating policy; (b) There are multiple constructive/operating alternatives (Table 1), which need to be comparatively analyzed.

The offline (model-based) numerical derivation of an optimal operating policy of a bioreactor (**FBR** in this work) is very commonly used in the engineering practice and achieves satisfactory results [13,14,16,17,18,31,32,33], being supported by several arguments: (i) The obtained problem solution is susceptible to an immediate implementation, as long as the obtained **FBR** operating policy is based on an adequate (validated) structured kinetic model; (ii) Optimal operation of bioreactors is a crucial engineering issue because it can lead to consistent economic benefits. Our results presented in the Section 5 prove it; (iii) Most of the kinetic models of moderate complexity are very flexible. Thus if, at a certain batch, significant inconsistencies are observed between the model-predicted bioreactor dynamics and the experimental evidence, the optimization stage is applied again, after an intermediate step (between batches) necessary to offline improve the model adequacy (model updating).

In the simplest **BR** operating alternative, biomass/substrates and other nutrients are initially added. Even if this bioreactor type has a low production capacity, it is commonly used for slow bioprocesses. Despite several drawbacks, **BRs** are highly flexible and easy to operate [31]. Due to these reasons, various **BRs** alternatives are often model-based verified, namely: (i) multi-objective **BR** optimization (that is of the batch time, substrate/biocatalyst initial load) [9,10,22,28,34]; (ii) a **SeqBR** optimization by including a model updating step based on acquired information from past batches (“tendency modeling”, not approached here) [4,11,12,17,19].

In contrast, other operating variants are more effective, in particular, those in which substrates/nutrients and/or biomass/enzymes are intermittently (**BRP**, [23]), or continuously (**FBR**, cyclic **FBR**, or **MASCR**) added during the batch by following a certain optimal policy (usually in-silico determined). A comparative discussion of the all mentioned bioreactor types and operating alternatives is provided by [35]. Recent advances reported better performances obtained by using multiple (48) fully automated mini-**FBR** operated in parallel [36].

A simpler but faster bioreactor optimization route was adopted in this paper, by comparing different operating alternatives of the **FBR** vs. **BR**.

However, the flexibility of **FBR** involves a more considerable modeling and computational effort to achieve a time-varying optimal operating policy (that is, an adjustable feeding policy with substrates/biomass) to obtain an efficient control of the biological process (see some examples in Table 1). The optimization problem often translates into a constrained multi-objective **FBR** optimization [14,16,19,20]. The superiority of **FBR** vs. **BR** should be model-based verified for every bioprocess case. Additionally, **FBRs** are more difficult to operate than **BRs**, as long as the time stepwise optimal feeding policy requires different feeding substrate solutions of different concentrations, and separate different cell cultures to be fed over the batch. This is the price paid for achieving **FBR** best performances. This need to previously prepare different cell cultures, and substrate stocks to be fed for every “time-arc” (that is a batch-time division in which the feeding is constant) is offset by the net higher productivity of **FBR** compared to those of **BR** as discussed in the below Section 5, and proved in the literature [14,16,23,37,38].

One very important application of **BRs/FBRs** is the industrial production of **mAbs** (of quasi-uniform characteristics), that is a molecule of major importance in medicine, by the so-called “Hybridoma technology” [39,40] by using antibody-secreting hybridoma cell cultures. However, the large-scale production of **mAbs** by using mammalian cells in optimized **BRs**, or **FBRs** (that is, with the continuous feeding of glucose (GLC) and glutamine (GLN) substrates and nutrients with variable concentrations/feeding rates; [33,40]) is limited due to the engineering problems associated with the model-based optimization, i.e., (i) the presence of multiple/contradictory objectives; (ii) the significant degree of uncertainty originating from multiple sources, and (iii) the unwanted decline in the cell viability that may occur during the batch. A short review of how some such engineering problems are approached is made by [5,9,17,21,41,42]. Also, process inhibition by several by-products (such as ammonium, and lactic acid, LAC, etc.) and/or hyper-osmotic stress related to nutrient feeds and base additions to control pH, all raise serious issues regarding the optimal operation of the batch-bioreactor. To reduce the LAC production, the use of adapted CHO (Chinese Hamster Ovary) cells was suggested by [43].

Other optimization routes of the bioreactor, not approached here, are focused on (i) the bioprocess model extension to account for the intracellular factors related to the cellular metabolic fluxes, or (ii) by improving the **FBR** operation by also accounting for the uncertainty in the model structure and its parameters, that is, the so-called “robust/stochastic” optimization, (see [44] denoted **LG17**, or [28,45,46]). Thus, several **FBR** optimization alternatives have been proposed in the literature, as follows: (a) offline one-time optimization not considering (nominal) or by considering (robust) the parameter uncertainty, with an offline derived process model, see **LG17**; or [47]; (b) optimization of the expected value of the objective function by considering the parametric uncertainty [46]; and (c) online **FBR** optimization, in which the feeding policy is adapted based on the online acquired information, and used to improve the model adequacy and the operating policy for the rest of the batch [21,29]. These formulations are solved using various nonlinear programming (NLP), mixed-integer (MINLP), or other solvers [48,49], due to the presence of a large number of constraints [11,20,21,30].

This paper aims to in-silico (offline) determine the optimal (time stepwise) feeding policy of a **FBR** with substrates, and with an immobilized hybridoma culture (on porous alginate beads) to maximize the production of **mAbs** over the batch by considering a small number of key control variables and an experimentally validated bioprocess dynamic model. The performance of the **FBR** in all checked alternatives is compared to those of a **BR**, in terms of raw material consumption and **mAbs** productivity. The study will illustrate: (i) in which cases the **FBR** is more effective than the **BR**, by using a simple operating policy given on every time-interval of equal length (i.e., over **N_div_** “time-arcs” in which the batch-time is divided; [37]), even if small **N_div_** values are checked (2 or 5 here, explained in Section 3.6), and (ii) the major influence of the control/decision variable searching ranges (set to be narrower or wider) on the optimal **FBR** efficiency. Multiple elements of novelty (i–iv) of the paper are underlined in Section 6 “Conclusions”. The in-silico analysis of the paper proved how such an optimal **FBR** operation is simpler to be implemented and more flexible (by presenting a larger number of degree of freedom, due to multiple control variables, in spite of a moderate small **N_div_**) compared to some optimal policies reported in the literature (e.g., [18]), that uses an exponential trajectory of the feeding liquid flow rate, and only the inlet levels of [GLC] and [GLN] as control variables, all being obtained by means of a hybrid deterministic (differential, intrinsic)–empiric (macroscopic) model.

As long as classical optimization of hybridoma **FBR** most of the time leads to some singular arc trajectories for biomass optimization, the advantage of the optimized **FBR** considering multiple control variables and a certain number of feed arcs that are not necessarily constant (that is, more subdivisions in the feed policy) becomes obvious, as this paper proves by using a detailed numerical analysis.

## 2. SBR Culture and Bioprocess Dynamics

A typical **BR** or **FBR** reactor is equipped with temperature, pressure (mechanical agitation, air sparger, pH, and dissolved oxygen (DO) control systems; see a simplified scheme in Figure 1). In the present study, the **BR** studied by **LG17** (that is [44]) was adapted to investigate a semi-continuous fed-batch bioreactor (**FBR**) operation mode, allowing for the addition of the substrate solution (with a variable feed flow rate), and of viable biomass (immobilized inside alginate porous beads [50,51,52]) during the batch according to optimal policies to be determined. This in-silico (model-based) analysis will allow comparing the **FBR** vs. **BR** performances under various operating alternatives.

To in-silico solve this problem, the use of both bioprocess, and bioreactor dynamic models is necessary. To support engineering calculations, several attempts have been reported in the literature to obtain adequate dynamic models to predict the key-species behavior in either **BR** or **FBR** systems as a function of the extracellular nutrient/metabolite concentrations, such as: (i) Models in reasonably reduced formulations but including the dominant factors involved in the optimization of the **mAbs** production [12,17,19,33], or **LG17**. (ii) More sophisticated bioprocess kinetic models that explicitly include intracellular factors related to the cellular metabolic fluxes [33,34,41,53], which are associated with the central carbon metabolism (**CCM**) [3]. However, such complex structured models of (ii) type are very difficult to identify, and their use for bioreactor optimization is very limited.

In the present study, the bioprocess kinetic model (denoted by **LGM**) adopted for **mAbs** production is that proposed by **LG17** ([44]). **LGM** is presented in Table 2, together with the associated rate constants. The hypotheses used to develop the kinetic expressions of **LGM** have been presented by **LG17** and are not further discussed in this paper. **LGM** was verified over extensive experiments performed by the authors. Besides, it is worth mentioning that the **LGM** is the reduced form of a previously published extended kinetic model (**EKM**), also experimentally validated by [33,54]. In the **LGM**, the essential terms account for the inhibition/limitation effects on the reaction rates due to the presence of various intra-/extracellular metabolites, substrates, or by-products in a similar way done by other reported models [34,41,55,56,57,58].

As presented in Table 2, the bioprocess **LGM** was included in the dynamic model of the **FBR** to describe the key-species dynamics during the batch. The bioreactor initial conditions and time stepwise values of the control variables will be further explored.

To not complicate the computational step, the adopted ideal model of **FBR** is a classical one [1], developed with the following simplifying hypotheses: (i) the operation is isothermal, iso-pH, and iso-DO; (ii) it is self-understood that nutrients (that is, compounds playing roles of sources of carbon, nitrogen, and phosphorus; [55]) are added initially and during the **FBR** operation, in recommended quantities, and of a C/N/P ratio of ca. 100/5/1 wt., together with an excess of aeration (pure oxygen, if necessary) for ensuring an optimally biomass maintenance, and any growth limitation due to such factors; (iii) the volume of the perfectly mixed liquid phase (with no concentration gradients) increases according to the liquid feed flow rate time-varying policy; (iv) the limits of the volumetric liquid feed flow rate (F_L,j_ in Table 2) are adjusted to ensure a maximum reactor content dilution of 10–25% relative to the initial liquid volume (V_L,0_) to not increase separation costs significantly; (v) there is negligible mass resistance for the transport of nutrients/substrates/products/oxygen into the liquid and in porous alginate beads; (vi) the substrates and solid carrier (less than one millimeter size), including the immobilized biomass, are initially added to the bioreactor, and then added during the batch according to an optimal feeding policy of the **FBR** to be determined; (vii) uniform solid particles are considered uniformly distributed in the homogeneous liquid phase, due perfect mixing conditions. In practice, perfect mixing conditions are never met. However, this is a usual hypothesis adopted for most of FBR models to not complicate them with hydrodynamic parameters difficult to be experimentally determined. A vigorous air sparging, and the use of a suitable mechanical mixing can support such a hypothesis.

The **FBR** is considered here with using immobilized biomass, for several reasons: (i) Its reported higher stability [50,51,52], and (ii) **FBR** operation with an online addition of the viable biomass (of variable concentration, coming from different cell cultures stocks prepared separately to be fed for every “time-arc”) should eventually be considered in our analysis (an option seldom described in the literature; refer to the optimal **MASCR** operation reported by [16]). To simplify the numerical analysis, the cell cultures stocks are assumed quasi-homogeneous and included as the Xv lump into the kinetic model.

The higher biomass fed concentration is used, the more vigorous aeration is employed (pure oxygen, if necessary). (iii) The used immobilized biomass is best suitable alternative for a variable (optimal) feeding of the **FBR** with biomass, because it is easy to dose [50,51,52].

From a mathematical point of view, the **FBR** dynamic model translates to a set of differential mass balances written for every considered species in the following general form:(1)$$\frac{{\mathrm{dC}}_{i}}{\mathrm{dt}}=\frac{F_{L}}{V_{L}}\left(C_{\mathrm{inlet},i}-C_{i}\right)\pm r_{i}\left(C\left(t\right),C_{0},k\right);C_{i,0}=C_{i}\left(t=0\right)$$
where species index i relates to X_v_, X_t_, GLC, GLN, LAC, AMM, and mAbs from the abbreviation list.

The reaction rate r_i_ expressions together with the associated rate constants and other details are given in Table 2. In Equation (1), C = the vector of the species concentrations; C_o_ = initial vector C (at time t = 0); k = the model rate constant vector. The reactor content dilution (determined by the increasing V_L_) is due to the variable F_L_(t) term.

In the model (1), the viable biomass (X_v_) dynamics is of the following generic form.
(2)$$\frac{{\mathrm{dX}}_{v}}{\mathrm{dt}}=X_{v}f\left(\xi \left(t\right)\right)-X_{v}g\left(\xi \left(t\right)\right),$$

The rate expressions f(ξ(t)) and g(ξ(t)) in (2) proposed in the literature are of a Monod-type and include terms accounting for the growth and death inhibition, respectively, caused by the concentration of the extracellular species, such as [GLC], [GLN], [LAC], [AMM], etc., see **LG17** and [33,41,56,57,58], or by intracellular metabolites related to the **CCM** [34]. The (X_v_) mass balance in the **LGM** of Table 2 is made fairly adequate by including the main influential species and terms, as pointed out and experimentally proved by **LG17**. Other authors [55,57] have proposed empirical kinetic forms for (2). Regardless of the approach, the initial/fed cell density, its quality, immobilization type, and medium characteristics clearly play the central role on the biomass dynamics, and **mAb** production yield.

To determine the species dynamics over the batch time (t_f_), the model (1) and (2) is solved with a proposed initial condition of C_i,0_ = C_i_ (t = 0), and using the best medium conditions shown in Table 3. Except for the control variables F_L,0_, [GLC]_0_, [GLN]_0_, and X_V,0_, (X_t,0_, = X_V,0_) whose initial values are to be determined by the optimization.

There are several reasons why the adopted **LGM** is preferred in our optimization analysis: (i) Its adequacy was experimentally validated and was reportedly fair; (ii) The **LGM** is the slightly reduced form of moderate complexity (19 rate constants in Table 2) of an **EKM** (of 31 rate constants, developed and validated by [33,54]). (iii) Even if the **EKM** is more complex by including differential mass balances of additional 8 intermediates, the dynamics of the 8 key state-variables (X_v_, X_t_, GLC, GLN, V_L_, LAC, AMM, and **mAbs**.) are fairly represented by both models. (iv) Being a reduced form of the **EKM**, the **LGM** is easier to be used, as it has fewer parameters (parsimonious principle). (v) Being simpler but adequate, the **LGM** is expected to offer interpretable results with less computational effort.

## 3. Formulation of the Bioreactor Optimization Problem

The optimal **FBR** operation will be compared with the reference **BR** used by **LG17** with a nominal **SPBR** (**S**et**P**oint of the **BR**) presented in Table 3, corresponding to the species dynamics plotted in Figure 2. The realized **BR** modest performance is of **Max[mAb]** = 1254.6 (mg/L) over a batch of t**_f_** = 100 h, as presented in Table 4 and Table 5 (**SPBR** line) In this **BR** simple operation, the substrates and immobilized biomass are initially loaded, and the **mAb** product is separated at the end of the batch. **FBR** operation is more complex, as below described.

### 3.1. Control Variables Selection

By analyzing the process model of Table 2, the natural option is to choose as control variables those that are related to reactor feeding with raw materials (GLC, GLN) and biomass (**X_v_**), whose concentrations play the major role in **mAbs** production. Additionally, the liquid feed flow rate **F_L_** will also be considered, being responsible for the reactor content dilution. Consequently, the selected control variables are as follows, including the inoculum size [34], for each time-arc (index “j”):(i).the continuously added liquid flow rate F_**L**,**j**_ (j = 1, …, N**_div_**);(ii).the time stepwise added [**GLC**]_**inlet**,**j**_; [**GLN**]_**inlet**,**j**_; [X_**v**_]_**inlet**,**j**_ (j = 1, …, N**_div_**);(iii).the **FBR** initial condition, that is, the initial liquid flow rate **F**_**L**,**0**_, and the initial substrates (as shown in Table 2), that is:[**GLC**]**_0_** = [**GLC**](t = 0) = [**GLC**]_**inlet**,**1**_;(3)
[**GLN**]**_0_** = [**GLN**](t = 0) = [**GLN**]**_inlet_**_,**1**_; [X**_v_**]**_0_** = [X**_v_**](t = 0) = [X**_v_**]**_inlet_**_,**1.**_(4)

### 3.2. Objective Function (Ω) Choice

By considering the mentioned control variables, the **FBR** optimization consists of determining the initial conditions, and the optimal feeding policy for every time-interval during the batch that leads to the maximization of the [**mAb**] produced during the batch, that is:Max **Ω**,(5)
where: **Ω** = Max [**mAb** (t)].

The **mAb**(**C**(t), C_o_, k, F_L_) (t) in (5) is model-based evaluated over the whole batch time
(t) ∈ [0, t**_f_** ],(6)

According to Equations (5) and (6), the objective function (**Ω**) consists in maximization of the [**mAb**] produced during the whole batch, and not only of the final [**mAb**]. That is because, once [**mAb**] reaches its maximum, the batch process can be stopped. In the present analysis, it happened that, for the derived optimum policies **SP1**–**SP3** (**S**et**P**oints of the **FBR** defined in Table 4), and for the **SPBR** (**S**et**P**oint of the **BR** presented in Table 3), the maximum [**mAb]** is reached at the batch end (Figure 2, Figure 3, Figure 4 and Figure 5).

### 3.3. Problem Constraints

(a)The **FBR** model (1)–(4) including the bioprocess kinetic model (Table 2);(b)The **FBR** initial condition, that is: [**GLC**]**_0_**, [**GLN**]**_0_**; F_L,0_; [X**_v_**]**_0_** = [X**_t_**]**_0_** (adopted);(c)The initial [**mAb**]**_0_**, [**AMM**]**_0_**, [**LAC**]**_0_**, adopted at values given in Table 3;(d)To limit the excessive consumption of raw materials, feasible searching ranges are imposed to the control/decision variable (with limits specified in Table 4), that is:

[**GLC**]**_inlet_**_,**min**_ ≤ [**GLC**]**_inlet_**_,**j**_ ≤ [**GLC**]**_inlet_**_,**max**_; F_L,min_ ≤ F_L,j_; F_L,0_ ≤ F_L,max_(7)

[**GLN**]_**inlet**,**min**_ ≤ [**GLN**]_**inlet**,**j**_ ≤ [**GLN**]_**inlet**,**max**_;[X**_v_**]_**inlet**,**min**_ ≤ [X**_v_**]_**inlet**,**j**_ ≤ [X**_v_**]_**inlet**,**max**_(8)

The imposed ranges for the control variables (tested wider or narrower) are related to not only the implementation possibilities, discussed by [12,17,52] and **LG17**, but also to economic reasons (minimum substrate consumption; effective control).

### 3.4. Searching (Control) Variables and Problem Formulation

To conclude, the **FBR** optimization problem consists in finding the optimal values of the initial and of the input levels of the four selected control variables, that is {[**GLC**]**_inlet_**_,**j**_; [**GLN**]**_inlet_**_,**j**_; [X**_v_**]**_inlet_**_,**j**_; F**_L_**_,**0**_; F**_L_**_,_**_j_**; j = 1, …, N**_div_**}, over N**_div_** time-intervals (“arcs”) of equal lengths Δt = t**_f_**/N**_div_** and under the specified operating constraints Equations (7) and (8), that maximize the chosen objective function Equations (5) and (6). In total, there are 4 × N**_div_** searching variables (i.e., 20 if N**_div_** = 5, or 8 if N**_div_** = 2). The time-intervals of equal lengths Δt = t**_f_**/N**_div_** are obtained by dividing the batch time into N**_div_** parts t**_j−1_** ≤ t ≤ t**_j_**, where t**_j_** = jΔt are switching points (where the reactor input is continuous and differentiable). Time-intervals are shown in the “Liquid volume dynamics:” row of Table 2. 

### 3.5. The Problem Solution

The problem solution will indicate the optimal running conditions of the FBR using the best medium conditions shown in Table 3, which maximize the objective Equations (5) and (6) in the presence of constraints Equations (7) and (8). As above formulated, the optimal operating policy will be given for every of time-intervals (of equal lengths) distributed throughout the batch-time.

### 3.6. (N_div_) and Operating Alternative Choice

The adopted **LGM** and the bioreactor dynamic model includes enough degrees of freedom to offer a wide range of **FBR** optimal operating alternatives that might be investigated (a–e) [not all being approached here], as follows:(a)by choosing unequal time-arcs, of lengths to be determined by the optimization rule;(b)by considering the whole batch time as an optimization variable;(c)by increasing the number of equal time-arcs (N**_div_**) to obtain a more “refined” and versatile **FBR** operating policy;(d)by considering the search min/max limits of the control variables as unknown (to be determined);(e)by feeding the bioreactor with solutions of uniform concentrations over a small/large number (N**_div_**) of time-arcs.

The alternative (d) is unlikely because it might indicate unrealistic results, such as unreachable [X_v_]**_inlet_**_,**j**_ high levels, or substrate high levels which can inhibit the bioprocess; or even [X_v_]**_inlet_**_,**j**_/substrates of too low levels that can stop the bioprocess. In our numerical analysis, carefully documented [X_v_]**_inlet_**_,**j**_, and substrates upper bounds were tested instead to ensure the practical implementation possibility of the optimal results. The alternative (e) is also not feasible, even if a larger (N**_div_**) will be used. That is because it is well-known that the variability in the feeding solutions over the batch time-arcs is the main degree-of-freedom used to obtain **FBR** optimal policies of superior quality [23,24]. By giving up to the variable concentrations in the feeding solutions, sub-optimally policies will be obtained.

Concerning the alternative (c), it is worth noting that, as (N**_div_**) increases, and the above alternatives (a,b) are considered as well, the necessary computational effort grows significantly (due to considerable increase in the number of searching variables), thus hindering the quick (real-time) implementation of the derived operating policy. Additionally, multiple optimal operating policies can exist for an over-parameterized constrained optimization problem with a high nonlinearity, increasing the difficulty to quickly locate a feasible globally optimal operating policy.

Additionally, as the (N**_div_**) increases, the operating policy is more difficult to implement since the optimal feeding policy requires a larger number of stocks with feeding substrate solutions of different concentrations, and different cell cultures stocks separately prepared to be fed for every time-arc of the **FBR** operation (a too expensive alternative). Also, the NLP problem is more difficult to solve because multiple optimal solutions may exist difficult to be discriminated and implemented. This is the case, for instance of an obtained optimal policy with a very high [X**_v_**]**_inlet_**_,**j**_ difficult to be ensured due to oxygenation limitations. Besides, **FBR** operation with using a larger number of small time-arcs (N**_div_**) can raise special operating problems when including PAT (Process Analytical Technology) tools [59].

A brief survey of the **FBR** (**SBR)** optimization literature [37,60] reveals that a small number (N**_div_**) <10 is commonly used due to the above-mentioned reasons. In fact, the present numerical analysis does not intend to exhaust all the possibilities of **FBR** optimization. Thus, an extended analysis of alternatives (a–e) of the **FBR** operation, or the influence of the parametric uncertainty deserves a separate investigation, which is beyond the scope of this paper.

To not complicate the computational analysis, only equal time-arcs have been tested, with the batch time t**_f_** = 100 h. Two alternatives were adopted: (i) (N**_div_**) = 5, in which equal time-arc-lengths of t**_f_**/(N**_div_**) = 20 h were used, and ii) (N**_div_**) = 2, with equal time-arc-lengths of t**_f_**/(N**_div_**) = 50 h. However, the search min/max limits of the control variables have been varied in each tested case.

### 3.7. The Used Solvers

The **mAb** time-evolution in Equation (5) is determined by solving the bioreactor dynamic model Equations (1)–(4) with a tested initial condition of C_j,0_ = C_j_ (t = 0), an imposed batch time t_f_, and the optimal medium conditions of Table 3. The dynamic model solution was obtained with enough precision by using the low-order stiff integrator (“ode23s”) of the MATLAB™.

Because the bioreactor/bioprocess nonlinear model and optimization objective Equations (1)–(8) are constrained, the problem translates into a nonlinear optimization problem (NLP). To obtain a global solution with enough precision, the multi-modal optimization solver MMA of [48,61] has been used, as being proved in previous works to be more effective than common/commercial algorithms. The computational time was reasonably short (minutes) using a common PC (Core-I7 processor), thus offering a quick implementation of the **FBR** optimal policy.

## 4. In-Silico Optimization Results

Further **FBR** optimization calculations will be made by considering the biomass with the characteristics of **LG17**. It is also assumed that the biomass retains its characteristics after immobilization, reflected by the same **LGM** rate constants of Table 2.

The offline (model-based) **FBR** optimization problem Equations (5) and (6) is realized gradually, by using various searching ranges to limit the raw materials consumption, and the liquid volume excessive increase. Three optimization alternatives have been checked (**SP** denotes a set-point), as follows:(1)**FBR-SP1.** For the adopted N**_div_** = 5, with equal time-arcs, and by using narrow search intervals for the control variables [**GLC**]**_inlet_**_,**j**_; [**GLN**]**_inlet_**_,**j**_; [X_v_]**_inlet_**_,**j**_; and F**_L_**_,**j**_ specified in Table 4 (in the **SP1** row), the obtained optimal operating policy **SP1** (for every time-arc) is presented in Table 4, together with the key-species dynamics in Figure 3. Final liquid volume is 1.27 × V**_L_**_,**0**_(2)**FBR-SP2.** For N**_div_** = 5 and equal time-arcs, but using wider search intervals for the (above mentioned) control variables, as specified in Table 4 (in the **SP2** row), the obtained optimal operating policy **SP2** of the control variables is presented for every time-arc in Table 4, together with the key-species dynamics in Figure 4. The final liquid volume is 1.1 × V_**L**__,**0**_.(3)**FBR-SP3.** For an adopted smaller N**_div_** = 2 with equal time-arcs, but using the same wide search intervals for the control variables as for the **SP2** case (**SP3** row in Table 4), the obtained optimal operating policy **SP3** of the control variables is presented for every time-arc in Table 4, together with the key-species dynamics during the batch in Figure 5. The final liquid volume is 1.1 × V**_L_**_,**0**_.

A comparison of substrate and biomass consumption among the derived optimal operating alternatives of **BR** and **FBR** is presented in Table 5 together with the realized performances, that is the follows setpoints:—**BR** (with nominal **SPBR** of Figure 2, and Table 4) [44], or—**BR** of [54], or—**FBR** (with set-points **SP1**, **SP2**, **SP3** of Figure 3, Figure 4 and Figure 5, Table 4),—**FBR** of [33].

The substrate consumption for FBR was evaluated with the formula: $\sum_{j=1}^{N_{\mathrm{div}}}F_{L}{\left[\mathrm{conc}.\mathrm{species}\right]}_{\mathrm{inlet},j}\Delta t_{j}$. Alternatively, for the **BR** case, raw materials consumption is based on the only initial load. The optimization alternative (d) (mentioned in the above Section 3.6), that is the choice of even wider search intervals for the control variables, was not approached, being not feasible due to the previously explained reasons. Carefully documented [X_v_]**_inlet_**_,**j**_ upper bounds were tested instead to ensure the practical implementation of the obtained optimal operating policy.

## 5. Results and Discussion

(I).In all the simulated alternatives, the **FBR** performance (in terms of produced mg **mAb**/L) is better than that realized by the **BR** (see the results summarized in Table 5), even if the overall batch time is the same (100 h), and a simple operating policy with equal time-arcs, in a small number (2–5) is used. The **FBR** productivity is up to 6× higher than that of the **BR**, while **FBR** is using fewer raw materials (Table 5). In Table 5, the **mAb** productivity is expressed in the absolute terms of Max [**mAb**] [mg/L]. Other indices, such as Max [**mAb**][mg/cells.h] can be used as well, by combining the data of Table 4 and Table 5. Being an intensive index, according to “Max [**mAb**][mg/cells.h]”, the **BR** appears more favorable because it uses less total [X**_v_**]. However, due to the large value of the product vs. the used biomass, such a poor advantage of the **BR** becomes negligible.(II).The study points out the major influence of the control variable setting ranges (narrower or wider) used by the optimization rule, on the obtained efficiency of the **FBR** optimal policy. More specifically, according to the results of Table 5, it turns out that:(IIa).The GLC consumption during **FBR** is ca. 1/2 for **SP1**, or 1/3 for **SP3** than that of the **BR** case. Similarly, fewer GLN was also consumed. The biomass (X_v_) consumption is roughly the same because of a smaller N**_div_** (**SP3**), or of narrower search ranges of control variables (**SP1**).(IIb).Non-uniform adding policy of (X_v_) and substrates is better in the **FBR** case (**SP2** and **SP3**; (Figure 4 and Figure 5d–g) than in the other cases, as the compensation of the biomass death is attempted (Figure 4 and Figure 5a,g), while maintaining a continuous increase in the produced **mAb**.(IIc).The price paid by the **FBR-SP2** to achieve the best performances compared to **BR-SPBR**, and **FBR-SP1**, or **FBR-SP3** is a higher consumption of raw materials, i.e., (vs. **BR**) of ca. 1.5× more GLC and GLN, and 10× more biomass.(IId).The **FBR-SP3** (X**_v_**_,**0**_, and **mAb** net productivity) policy appears to be somehow intermediate between the **FBR-SP2** and **BR-SPBR**. Compared to the **SPBR**, the raw material consumptions are smaller, but the realized **FBR** performances are better, because an operation with N**_div_** = 2 is more versatile than that of the **BR** with the initial load being the only optimization option.(IIe).The used biomass is generally higher in the **FBR** cases compared to those of **BR** (Table 5). Thus, the used (X**_v_**) is roughly the same for **SP3** (N**_div_** = 2), but **2×** for **FBR-SP1**, or 10× for **FBR-SP2**.(III).The GLC consumption for **FBR** operating case depends on the inlet GLC policy (the [**GLC**]**_inlet_**_,**j**_ term in Table 2), and on the used control variable dynamics (X**_v_**, GLN).(IV).A comparison indicates that the GLC dynamics of the **SP1** vs. **SP2** of the **FBR** (with N**_div_** = 5) is depending on not only the searching interval chosen for the control variable [**GLC**]**_inlet_**_,**j**_, but also on the other species dynamics. Thus, if one compares the (X**_v_**) plots of **SP1** in Figure 3 to that of **FBR-SP2** in Figure 4, it is easy to observe and explain that when (X_v_) is high, GLC consumption is also high, in spite of a larger inlet [GLC]. This clearly shows that the **FBR** optimization must consider all variables simultaneously.(V).In the **BR-SPBR** case of **LG17**, a less flexible feeding explains its modest **mAb** productivity. The **BR** species dynamics (in Figure 2) is comparable to those of **FBR-SP3** (N**_div_** = 2; in Figure 5), that is, GLC is quickly consumed during the first half of the batch, and the biomass displays a pronounced peak in the first half of the batch.In the end, it is worth mentioning that the present numerical engineering analysis presents multiple elements of novelty as briefly mentioned in Section 6 “Conclusions”.(VI).Our results prove the multiple advantages obtained when using **FBR** operated with multiple control variables following very versatile optimal feeding policies consisting in time stepwise variable of: (i) the feeding liquid flow rate, (ii) the added [GLC], (iii) the added [GLN], and (iv) the added [X_v_] over the batch.The in-silico analysis of the paper proved how such an optimal **FBR** operation is leading to quick results, easy to interpret and to implement, being more flexible and effective due to a larger number of degree of freedom (coming from the multiple control variables, and from their variable time stepwise policy), in spite of an economically advantageous small number of employed time-arcs (N**_div_**) compared to some optimal policies of a similar **FBR** reported in the literature. For instance, [18] uses only an exponential trajectory of the feeding liquid flow rate, and only the inlet levels of [GLC] and [GLN] as control variables, all being obtained by using a hybrid deterministic (differential, intrinsic)–empiric (macroscopic) model.(VII).The present in-silico (model-based) analysis have not been experimentally validated. However, as long as various forms of the used **LGM** were experimentally validated in a multiple and independent manner by **LG17**, and by [33,54] (Section 2), the results obtained by our numerical analysis shows sufficient credibility from the engineering point of view, from the following reasons:(a)Even if an experimental validation of the derived optimal policy **FBR-SP2** policy is missing, our paper presents a very strong engineering value by exemplifying, in a relatively simple manner, a numerical procedure (process model-based) that can be used to solve similar complex optimization problems of **FBR**.(b)Such an **FBR** optimal control rule is possible because most kinetic models of moderate complexity are very flexible. Thus, if significant inconsistencies are observed between the model-predicted bioreactor dynamics (e.g., optimal policy **SP2** in the present case) and the experimental data, then the optimization stage is applied again by using the same rule, but after performing an intermediate numerical-analysis step (between batches) necessary to improve the model adequacy (the so-called “model updating” based on the online measurements).(VIII).As displayed in Figure 2, Figure 3, Figure 4 and Figure 5 inhibition given by the increasing LAC, and AMM by-product concentration cannot be diminished by simple manipulations of the chosen control variables, even if the derived operating policy is an optimal one. However, the adopted kinetic model is able to fairly predict the dynamics of these inhibitory species. But the adverse side effects, such as a low pH, or a hyper-osmotic stress (due to the nutrient feeds and base additions to control pH) cannot be avoided by the above-used engineering (model-based) rules. As revealed in the literature, “biological” solutions are used instead to cope with such a problem. For instance, to reduce the LAC production, the use of adapted CHO (Chinese Hamster Ovary) cells can be an alternative [43].(IX).The used time-arcs of constant control variables are of 20 h (for SP1, SP2), and of 50 h (for SP3). Such an operation cannot raise special operating problems for a FBR, with also including PAT (Process analytical technology) tools. This is an additional argument not using a larger number of small time-arcs (N_div_, see Section 3.6).(X).The comparison in Table 5 of the obtained optimal policies **FBR-SP2** and **FBR-SP3** with those from the literature for a **BR** [44,54], or a **FBR** [33] operated with the same cell culture, indicates better performances despite a longer batch time, and a larger number of substrate feeding solutions of those in the literature.

## 6. Conclusions

To conclude, the optimized **FBR** operation with a time stepwise control of the feeding policy reported better performances than the simple **BR** operation due to its higher flexibility in using biomass and substrates, even if a small number of equal time-arcs is used. The **FBR** major drawback is coming from its difficult operation, as long as the time stepwise optimal feeding policy requires different feeding substrate solution stocks of different concentrations, and separate different cell cultures stocks to be fed over the batch. This is the price paid for achieving improved **FBR** performances. However, its reported excellent performances fully justify the extra investment in implementing the **FBR** optimal operating policy. An economic global evaluation accounting for the product/raw material value can give a more accurate answer to such a sensitive issue.

The present optimization analysis proves its worth by including multiple elements of novelty, as follows: (i) An optimally operated **FBR** with a small number of time-arcs (below 10) and using wider but feasible ranges for setting the control variables can lead to high performances of the bioreactor. (ii) The major role played by the variable feeding with the viable biomass, leading to consider (X**_v_**) as a control variable during **FBR** optimization (an option seldom discussed in the literature). (iii) The major influence of the control variable searching ranges (set to be narrower or wider) on the resulted efficiency of the **FBR** optimal policy. (iv) The model-based optimal operation of bioreactors is a very important engineering issue because it can lead to consistent economic benefits, as proved by the results presented in Table 5.

The present in-silico engineering analysis preserves its value by offering a useful example and comparative results for further engineering applications seeking for optimization of a **FBR** used to conduct complex bioprocesses.

The paper proves in a simple yet suggestive way how a lumped but adequately detailed dynamic model of a bioprocess can successfully support in-silico engineering evaluations by aiming to optimize the **FBR** operation if the nano-scale cell metabolism (including metabolic by-products) is somehow reflected by the bioreactor macro-scale dynamic model. In such a way, the considerable experimental and computational effort required to develop hybrid cell/bioreactor models of good quality is fully justified through the benefits of subsequent model-based engineering analyses that assist the **FBR** optimal operation.

## Figures and Tables

**Figure 1 molecules-25-05648-f001:**
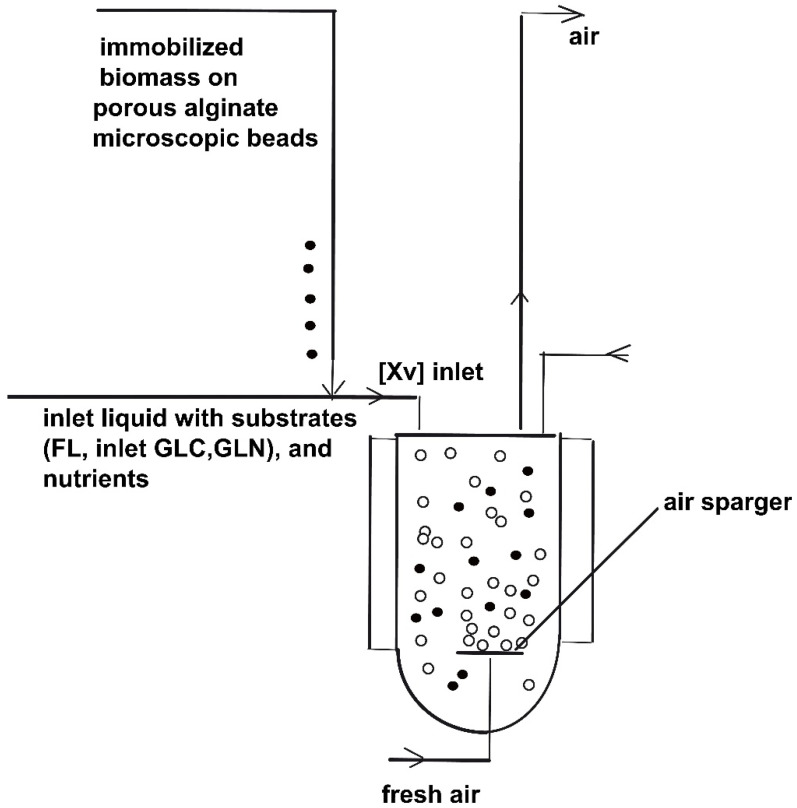
The fed-batch bioreactor (**FBR**) simplified scheme. The bioreactor is operated in a fed-batch mode, with time stepwise continuous feeding addition of substrates, nutrients, and immobilized biomass (under one millimeter size alginate beads) at levels to be determined by optimization for each “time-arc”.

**Figure 2 molecules-25-05648-f002:**
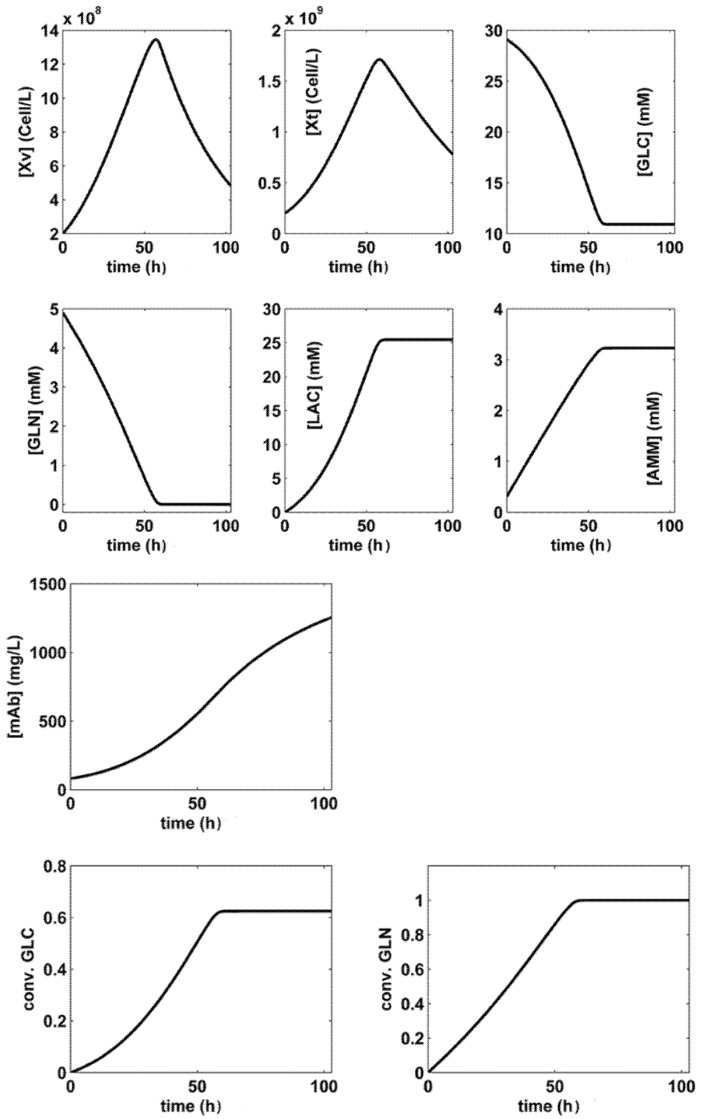
The simulated **SPBR** (**S**et**P**oint of the **BR** given in Table 3). Key species dynamics (that is time-trajectories) are generated by using the **LGM** of **LG17** [44].

**Figure 3 molecules-25-05648-f003:**
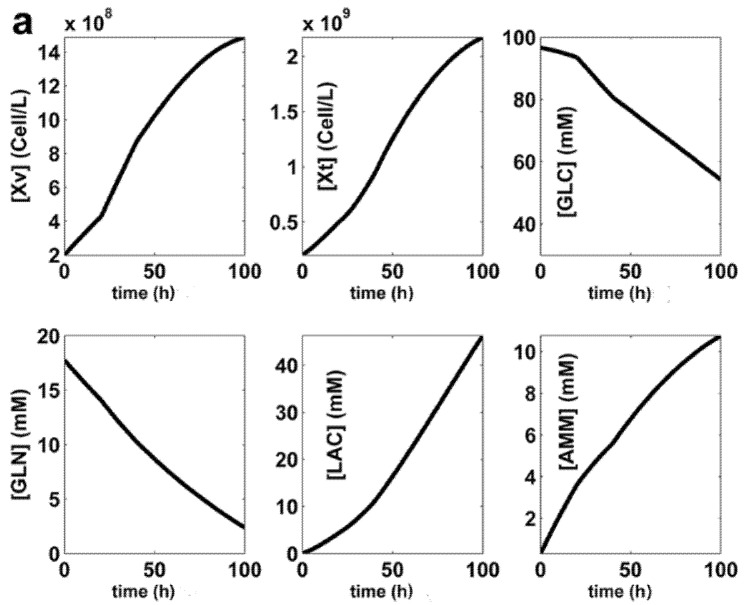
The simulated **SP1** (the optimum policy **S**et**P**oint no.1 of the **FBR**, defined in Table 4). The plots refer to the dynamics of the key species (**a**,**b**), and of the liquid volume (**c**). The plots (**d**–**g**) refer to the time stepwise optimal policy of the control variables **[GLC]_inlet_**(t) (**d**); **F_L_**(t) (**e**); **[GLN]_inlet_**(t) (**f**), and **[X_v_]_inlet_**(t) (**g**) for the approached **FBR** (the running details are given in Table 4). Species trajectories are generated by using the **LGM** model (Table 2).

**Figure 4 molecules-25-05648-f004:**
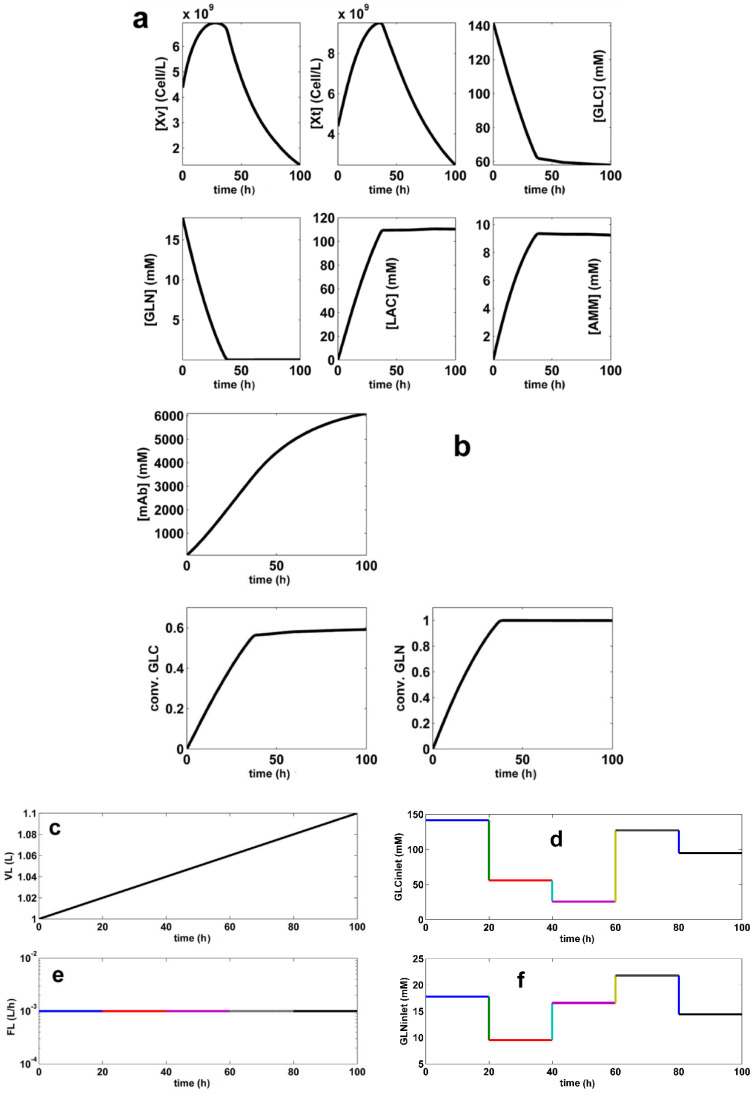
The simulated **SP2** (the derived optimum policy **S**et**P**oint no.2 of the **FBR**, defined in Table 4). The plots refer to the dynamics of the key species (**a**,**b**), and of the liquid volume (**c**). The plots (**d**–**g**) refer to the time stepwise optimal policy of the control variables **[GLC]_inlet_**(t) (**d**); **F_L_**(t) (**e**); **[GLN]_inlet_**(t) (**f)**, and **[X_v_]_inlet_**(t) (**g**) for the **FBR** approached in this paper (running details are given in Table 4). Species trajectories are generated by using the of **LGM** model (Table 2).

**Figure 5 molecules-25-05648-f005:**
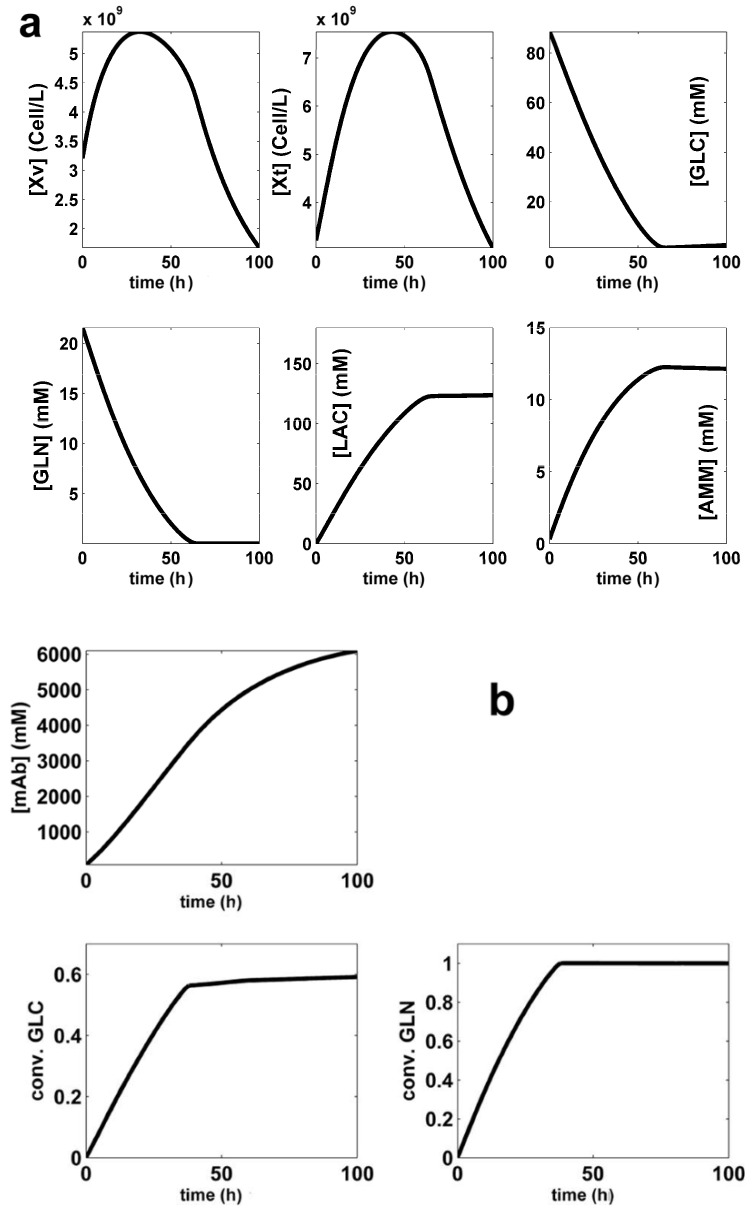
The simulated **SP3** (the derived optimum policy **S**et**P**oint no.3 of the **FBR**, defined in Table 4). The plots refer to the dynamics of the key species (**a**,**b**), and of the liquid volume (**c**). The plots (**d**,**g**) refer to the time stepwise optimal policy of the control variables **[GLC]_inlet_**(*t*) (**d**); F_L_(t) (**e**); **[GLN]_inlet_**(t) (**f**), and [X_v_]_inlet_(t) (**g**) for the **FBR** approached in this paper (running details are given in Table 4). Species trajectories are generated by using the of **LGM** model (Table 2).

**Table 1 molecules-25-05648-t001:** The main constructive and operating alternatives of bioreactors [1,4,5].

Reactor Type	Notation [Examples]	Operation; Modeling Hypotheses
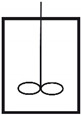	(i) simple **B**atch **R**eactor (**BR**) Examples: [9,10]	isothermal, iso-pH, and iso-DO (air sparger); perfectly mixed liquid phase (with no concentration gradients, by using mechanical agitation). Reactants/biomass added at the beginning of the batch only.
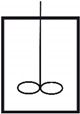	(ii) **SeqBR** (**S**equential **B**atch-to-batch **R**eactor) Examples: [11,12]	Ibidem. Reactants and/or biomass added at the beginning of each batch, in optimized amounts (to be determined)
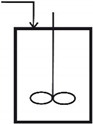	(iii) **S**emi-**B**atch (fed-batch) **R**eactor (**SBR** or **FBR**). Examples: [13,14,15,16,17,18,19,20,21]	Ibidem. Substrates/biomass/supplements added during the batch by following a certain (optimal) policy (to be determined)
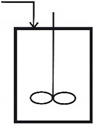	(iv) **BRP** (**B**atch **R**eactor with intermittent/**P**ulse-like additions of biocatalyst/substrates). Examples: [22,23,24,25]	Ibidem. Reactants and/or biomass added during the batch in a **P**ulse-like additions of equal/uneven solution volumes, with a certain frequency (to be determined)
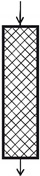	(v) continuously operated packed-bed columns, **FXBR** (**F**i**X**ed-**B**ed continuous bio**R**eactor) Examples: [23]	immobilized enzyme on a porous support packed in columns; continuous fed of the substrate/nutrient solution; continuous solution output; various aeration alternatives. Model hypotheses: isothermal, ideal plug-flow reactor of constant volume, with model dynamic terms allowing simulating transient operating conditions and the continuous enzyme/biomass deactivation
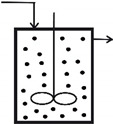	(vi) **MA(S)CR** (**M**echanically **A**gitated (**S**emi-)Continuous **R**eactor) (three-phases). Examples: [14]	immobilized enzyme on porous support suspended in the mechanically agitated bioreactor, with sparged gas (air); continuous fed of the substrate/nutrient solution, with/without continuous evacuation; Model hypotheses: isothermal, ideal perfectly mixed liquid phase (with no concentration gradients, by using mechanical agitation, aeration), with model dynamic terms allowing simulating transient operating conditions and the continuous enzyme/biomass deactivation. Substrates/biomass can be added with a constant/variable feed flow rate (to be determined).

**Table 2 molecules-25-05648-t002:** Key-species mass balances in the fed-batch bioreactor **FBR** model, including the bioprocess kinetic model **LGM** of **LG17**, together with the associated rate constants. Note: the ideal model below (of homogeneous liquid composition), is neglecting the mass transport resistance in the porous beads. Rate constants have been estimated by [44] from experiments that use the mammalian hybridoma cell culture of [54].

Species	Parameters	Remarks
**Biomass balance:** **Viable biomass balance:** $\frac{dX_{v}}{dt}=\frac{F_{L}}{V_{L}}\left(X_{v,inlet,j}-X_{v}\right)+\left(\mu -\mu_{d}\right)X_{v}$ $X_{v,inlet,j}$ = control variable; *j* = 1, …, $N_{div}$ time stepwise values to be optimized; $X_{v,0}$ = $X_{v,inlet,1}$; $X_{v,inlet}\left(t\right)$ to be optimized; **Total biomass balance:** $\frac{dX_{t}}{dt}=\frac{F_{L}}{V_{L}}\left(X_{t,inlet}-X_{v}\right)+\mu X_{v}-K_{lysis}\left(X_{t}-X_{v}\right)$ $X_{t,inlet}=0$ (adopted); $X_{t,0}$ = $X_{v,0}$ where: $\mu =\mu_{\max}\left(\frac{\left[GLC\right]}{K_{glc}+\left[GLC\right]}\right)\left(\frac{\left[GLN\right]}{K_{gln}+\left[GLN\right]}\right)$ $\left(\frac{KI_{lac}}{KI_{lac}+\left[LAC\right]}\right)\left(\frac{KI_{amm}}{KI_{amm}+\left[AMM\right]}\right)$ $\mu_{d}=\frac{\mu_{d,\max}}{1+{\left(K_{d,amm}/\left[AMM\right]\right)}^{2}}$	$K_{lysis}$ = 0.0551 h^−1^ $\mu_{\max}$ = 0.058 h^−1^ $K_{glc}$ = 0.75 mM $K_{gln}$ = 0.075 mM $KI_{lac}$ = 172 mM $KI_{amm}$ = 28.5 mM $\mu_{d,\max}$ = 0.03 h^−1^ $K_{d,amm}$ = 1.76 mM	**LGM**
**Balance of other species:** $\frac{d\left[GLC\right]}{dt}=\frac{F_{L}}{V_{L}}\left({\left[GLC\right]}_{inlet,j}-\left[GLC\right]\right)-Q_{glc}X_{v}$, ${\left[GLC\right]}_{inlet,j}$ = control variable; *j* = 1, …, $N_{div}$ time stepwise values to be optimized; ${\left[GLC\right]}_{0}$ = ${\left[GLC\right]}_{inlet,1}$; ${\left[GLC\right]}_{inlet}\left(t\right)$ to be optimized where: $Q_{glc}=\frac{\mu }{Y_{x,glc}}+m_{glc}$	$Y_{x,glc}$ = 1.06 × 10^8^ cell/mmol $m_{glc}$ = 4.85 × 10^−14^ mmol/cell.h	**LGM**
$\frac{d\left[GLN\right]}{dt}=\frac{F_{L}}{V_{L}}\left({\left[GLN\right]}_{inlet,j}-\left[GLN\right]\right)-Q_{g_{ln}}X_{v}-K_{d,g_{ln}}\left[GLN\right]$, ${\left[GLN\right]}_{inlet,j}$ = control variable; *j* = 1, …, $N_{div}$ time stepwise values to be optimized; ${\left[GLN\right]}_{0}$ = ${\left[GLN\right]}_{inlet,1}$; ${\left[GLN\right]}_{inlet}\left(t\right)$ to be optimized, where: $Q_{g_{ln}}=\frac{\mu }{Y_{x,g_{ln}}}+m_{g_{ln}}$ $K_{d,g_{ln}}$ = 0.0096 h^−1^; $Y_{x,g_{ln}}$ = 5.57 × 10^8^ cell/mmol $m_{g_{ln}}$ = $\alpha_{1}$ = −0.00067 mmol/cell.h		**LGM**
Liquid volume dynamics: $\frac{dV_{L}}{dt}=F_{L,j};\;F_{L}=\{\begin{array}{c} F_{L,0}\;\mathrm{if}\;0\leq t<T1\\ F_{L,1}\;\mathrm{if}\;T1\leq t<T2\\ F_{L,2}\;\mathrm{if}\;T2\leq t<T3\\ F_{L,3}\;\mathrm{if}\;T3\leq t<T4\\ F_{L,4}\;\mathrm{if}\;T4\leq t<t_{f} \end{array}$, (a) For the adopted $N_{div}$ = 5, the *j* = 1, …, $N_{div}$ time-arcs switching points are: T1 = 20 h.; T2 = 40 h.; T3 = 60 h.; T4 = 80 h.; $t_{f}$ = 100 h., where $F_{L,0}-F_{L,4}$ time stepwise values are to be determined together with the other control variables to ensure an optimal **FBR** operation; (b) For the adopted $N_{div}$ = 2, the *j* = 1, …, $N_{div}$ time-arcs switching points are: T1 = 50 h.; $t_{f}$ = 100 h., with $F_{L}=\{\begin{array}{c} F_{L,0}\;\mathrm{if}\;0\leq t<T1\\ F_{L,1}\;\mathrm{if}\;T1\leq t<t_{f} \end{array}$ where $F_{L,0}-F_{L,1}$ time stepwise values are to be determined together with the other control variables to ensure an optimal **FBR** operation;		**This paper**
$\frac{d\left[LAC\right]}{dt}=\frac{F_{L}}{V_{L}}\left({\left[LAC\right]}_{inlet}-\left[LAC\right]\right)+Q_{lac}X_{v}$, ${\left[LAC\right]}_{inlet}={\left[LAC\right]}_{0}$ = 0 where: $Q_{lac}=Y_{lac,glc}Q_{glc}$	$Y_{lac,glc}$ = 1.4 L	**LGM**
$\frac{d\left[AMM\right]}{dt}=\frac{F_{L}}{V_{L}}\left({\left[AMM\right]}_{inlet}-\left[AMM\right]\right)+Q_{amm}X_{v}+K_{d,gln}\left[GLN\right]$ ${\left[AMM\right]}_{inlet}$ = 0; ${\left[AMM\right]}_{0}$ = 0.31 mM where: $Q_{amm}=Y_{amm,g_{ln}}Q_{g_{ln}}$ $K_{d,g_{ln}}$= 0.0096 h^−1^; $Y_{amm,g_{ln}}$ = 0.427 L		**LGM**
$\frac{d\left[mAb\right]}{dt}=\frac{F_{L}}{V_{L}}\left({\left[mAb\right]}_{inlet}-\left[mAb\right]\right)+\left(2-\gamma \;\mu \right)\;\lambda \;X_{v}$ ${\left[mAb\right]}_{inlet}$ = 0; ${\left[mAb\right]}_{0}$ = 80.6 mg/L	γ = 0.1 h λ = 7.21 × 10^−9^ mg/(cell·h)	**LGM**

**Table 3 molecules-25-05648-t003:** The nominal operating conditions (**SPBR**) of **LG17** for the batch bioreactor **BR** with suspended mammalian hybridoma cell culture.

Parameter	Nominal Value	Remarks (*)
Total cell initial density (X_t,0_)	2 × 10^8^ Cell/L	Ref. to reactor-lq.
Viable cell initial density (X_V,0_)	2 × 10^8^ Cell/L	Ref. to reactor-lq.
Glucose initial concentration, [GLC]_0_	29.1, mM	
Glutamine initial concentration, [GLN]_0_	4.9, mM	
Lactate initial concentration, [LAC]_0_	0, mM	
Ammonia initial concentration, [AMM]_0_	0.31, mM	
Monoclonal antibody initial concentration, [mAb]_0_	80.6, mg/L	Ref. to reactor-lq.
Temperature	35–37 °C	[42]
pH (buffer, using CO_2_ injection)	7	See an optimal policy given by [42]
Aeration in excess, nutrients in sufficient amounts		[42,55]
Initial volume of the liquid in the bioreactor (V_L,0_)	1 L	**LGM**
Batch time (t_f_)	approx. 100 h.	**LGM**

(*) Ref. to reactor-lq. = Value relative to the liquid volume of the reactor.

**Table 4 molecules-25-05648-t004:** Model-based derived optimal operating policies for the approached **FBR** comparatively to performances of **LG17** batch bioreactor (**BR**) with immobilized mammalian hybridoma cell culture. Biomass and medium characteristics are those given by **LG17** in Table 3. The larger number of displayed digits comes from the numerical simulations.

Reactor SP	Searching Policy	Control Variables					Obs.
**SPBR** (**BR**) Optimal Values	Sensitivity Analysis (Exhaustive)	Initial Values of the **BR** Content					
		[GLC], mM		[GLN], mM	X_v,0_ = X_t,0_, Cell/L	Max [mAb](t), (mg/L)	
		29.1		4.9	2 × 10^8^	1254.6	LGM
**SP1**^(a)^ (**FBR**) Optimal values ^(d)^	Searching variables	F_L_, L/h.	[GLC]_inlet_, mM	[GLN]_inlet_, mM	X_v,inlet_ Cell/L		This paper
	Searching ranges	(10^−4^–10^−2^)	(25–100)	(5–25)	(2 × 10^8^–2 × 10^9^)		
	Multi-dimensional optimization	Inlet optimal values of the FBR control variables					
		F_L_, ^(b,c)^, L/h.	[GLC]_inlet_ mM	[GLN]_inlet_ mM	X_v,inlet_ Cell/L	Max [mAb](t), (mg/L)	
	Time interval (0, 20) h.	10^−3^	96.62	17.75	2 × 10^8^		This paper
	Time interval (20, 40) h.	9.55 × 10^−3^	45.13	9.52	1.7 × 10^9^		
	Time interval (40, 60) h.	10^−3^	26.99	16.58	1.62 × 10^9^		
	Time interval (60, 80) h.	10^−3^	87.80	21.77	1.33 × 10^9^		
	Time interval (80, 100) h	10^−3^	68.42	14.43	5.74 × 10^8^		
		Optimal value of Max [mAb](t), (mg/L)				1351.3	This paper
**SP2**^(a)^(**FBR**) Optimal values ^(d)^	Searching variables	F_L_, L/h.	[GLC]_inlet_, mM	[GLN]_inlet_, mM	X_v,inlet_, Cell/L		This paper
	Searching ranges	(10^−4^–5 × 10^−2^)	(25–150)	(5–25)	(2 × 10^8^–5 × 10^9^)		
	Multi-dimensional optimization	Inlet optimal values of the FBR control variables					
		F_L_ ^(b,c)^, L/h.	[GLC]_inlet_ mM	[GLN]_inlet_ mM	X_v,inlet_, Cell/L	Max [mAb](t), (mg/L)	
	Time interval (0, 20) h.	10^−3^	141.63	17.76	4.38 × 10^9^		This paper
	Time interval (20, 40) h.	10^−3^	55.81	9.52	4.20 × 10^9^		
	Time interval (40, 60) h.	10^−3^	25.60	16.58	3.98 × 10^9^		
	Time interval (60, 80) h.	10^−3^	126.92	21.77	3.21 × 10^9^		
	Time interval (80, 100) h	10^−3^	94.62	14.43	1.20 × 10^9^		
		Optimal value of Max [mAb](t), (mg/L)				6098.4	This paper
**SP3**^(a)^ (**FBR**) Optimal values ^(d)^	Searching variables	F_L_, L/h.	[GLC]_inlet_, mM	[GLN]_inlet_, mM	X_v,inlet_, Cell/L		This paper
	Searching ranges	(10^−4^–5 × 10^−2^)	(25–150)	(5–25)	(2 × 10^8^–5 × 10^9^)		
	Multi-dimensional optimization	Inlet optimal values of the FBR control variables					
		F_L_ ^(b,c)^ L/h.	[GLC]_inlet_ mM	[GLN]_inlet_ mM	X_v,inlet_, Cell/L	Max [mAb](t), (mg/L)	
	Time interval (0, 50) h.	10^−3^	88.65	21.58	3.21 × 10^9^		This paper
	Time interval (50, 100) h.	10^−3^	137.97	20.65	1.2 × 10^9^		
		Optimal value of Max [mAb](t), (mg/L)				5700.1	This paper

^(a)^ Time step -wise optimal operating policy of the **FBR** (plotted in Figure 3, Figure 4 and Figure 5). ^(b)^ The minimum feed flow rate of the inlet liquid **F_L_** was set to be around 10%V_L,0_/t_f_ = 10^−3^ L/h., or even below, to avoid excessive dilution of the bioreactor content [23]. The **F_L_**(t) policy during the batch is to be optimized, being adjusted so that the final dilution of reactor content doesn’t exceed 10–25% of the initial liquid volume. ^(c)^ The initial liquid volume in the **FBR** (V_L,0_), was adopted to 1 L, as for the **BR** case, that is 5× larger than that of [33]. ^(d)^ The optimal values refer to the inlet levels of the control variables for every time-arc, including the adjustable immobilized inlet X_t,inlet_. In the X_t_ case, the initial value is X_t,0_ = X_v,0_. During the batch, the inlet X_t,inlet_ is taken 0.

**Table 5 molecules-25-05648-t005:** Substrate and biomass consumption, and realized performances by **BR**, and by the optimally operated **FBR**. The **BR** initial load, and the **FBR** optimal feeding policy are presented in Table 4. The equal time-arcs of the **FBR** are of 20 h. each for **SP1** and **SP2**, and of 50 h. each for **SP3**. The **BR**, and **FBR** initial volume is of 1 L. The batch time is 100 h. in all cases, excepting for the last two lines.

Bioreactor Operation			Raw Material Consumption ^(b)^			Reactor Performance Max [mAb](t), ^(b)^		FBR Dilution
Type	N_div_	Set-Point	Consumed GLC (mmoles)	Consumed GLN (mmoles)	X_v,0_ (cells) ^(c)^	(mg/L)	(mg/cells/h)	(%) ^(a)^
**BR**	1	Nominal [44] **SPBR** ^(d)^	29.1	4.9	2 × 10^8^	1254	6.3 × 10^−8^	0
**FBR**	5	Optimal **SP1**	14.22	3.23	ca.4 × 10^8^	1351	3.4 × 10^−8^	27
**FBR**	5	Optimal **SP2**	44.46	8.00	1.7 × 10^9^	6098	3.6 × 10^−8^	10
**FBR**	2	Optimal **SP3**	11.33	2.11	2.2 × 10^8^	5700	2.6 × 10^−7^	10
**BR**	1	[54] ^(e)^			1–5 × 10^8^	~1100		0
**FBR**	7–13	[33] ^(f)^			2 × (10^8^–10^9^)	~2400		?

^(a)^ Referring to the reactor liquid initial volume. ^(b)^ The larger number of displayed digits comes from the numerical simulations. ^(c)^ Referred to the **FBR** initial volume of 1 L (Table 3). ^(d)^ The **BR** nominal set-point (Table 3) of [44]. ^(e)^ The same cell culture, 120 h batch time (results from experimental plots). ^(f)^ The same cell culture, 168 h batch time (results from experimental plots); Initial [**GLC**] = 5.5–25 mM; Initial [**GLN**] = 3.74 mM; F_**L**,**j**_ = 0–12.5 mL/h; V_0_ = 0.2 L.

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
