# Peer review of "Model-Based Optimization of a Fed-Batch Bioreactor for mAb Production Using a Hybridoma Cell Culture"

_molecules, 2020, doi:10.3390/molecules25235648_

Round 1

Reviewer 1 Report

The author has made many of the changes I suggested.However, I still believe that feeding the reactor with previous cultures of hybridomas in every time-arc during the operation of the fed-batch reduces the interest of the study. I am also convinced that this type of article does not fit into the journal's Organic Chemistry section.

Author Response

see the point-by-point answers included in the attached file "reply-reviewer-1"

Reviewer 2 Report

Please see the attached .pdf file.

Author Response

see the point-by-point answers included in the attached file "reply-reviewer-2"

Reviewer 3 Report

In this paper the Author describes a numerical model that would allow optimized production of monoclonal antibodies using hybridoma cells in a fed-batch bioreactor. The aim of the manuscript is to determine the optimal feeding policy using immobilized hybridoma culture to maximize mAb production. The paper concludes that, despite having a more complex operation, a fed-batch bioreactor model has superior performance compared to batch reactors.

The manuscript is well written, and the conclusions are supported by the modelling.

I only have minor comments:

The drawing shown in Fig. 1 could be improved by using more adequate graphics software

Since in practice perfect mixing conditions are unlikely to be reached, the author could add some comments as to the degree sub-optimal mixing might affect mAb production.

Please revise English language usage (minor changes)

Round 2

Reviewer 2 Report

The author has sufficiently addressed my concerns and comments they received regarding the original version of the paper with relevant and thorough replies, making significant improvements to the manuscript, thus further reinforcing the importance of the findings presented and the impact of the work.

Therefore, I encourage the immediate publishing of this paper as an appreciated contribution to the topic of mAb production using hybridoma cell cultures.